# Gaussian Splatting to Real World Flight Navigation Transfer with Liquid Networks

**Alex Quach**[*]
aquach@mit.edu

**Makram Chahine**[*]
chahine@mit.edu

**Alexander Amini**
amini@mit.edu

**Ramin Hasani**
rhasani@csail.mit.edu

**Daniela Rus**
rus@csail.mit.edu

[*] equal contribution
CSAIL, MIT
Project Website

**Abstract:** Simulators are powerful tools for autonomous robot learning as they offer scalable data generation, flexible design, and optimization of trajectories. However, transferring behavior learned from simulation data into the real world proves to be difficult, usually mitigated with compute-heavy domain randomization methods or further model fine-tuning. We present a method to improve generalization and robustness to distribution shifts in sim-to-real visual quadrotor navigation tasks. To this end, we first build a simulator by integrating Gaussian Splatting with quadrotor flight dynamics, and then, train robust navigation policies using Liquid neural networks. In this way, we obtain a full-stack imitation learning protocol that combines advances in 3D Gaussian splatting radiance field rendering, crafty programming of expert demonstration training data, and the task understanding capabilities of Liquid networks. Through a series of quantitative flight tests, we demonstrate the robust transfer of navigation skills learned in a single simulation scene directly to the real world. We further show the ability to maintain performance beyond the training environment under drastic distribution and physical environment changes. Our learned Liquid policies, trained on single target manoeuvres curated from a photorealistic simulated indoor flight only, generalize to multi-step hikes onboard a real hardware platform outdoors.

**Keywords:** End-to-end learning, Gaussian Splatting, Sim-to-real transfer

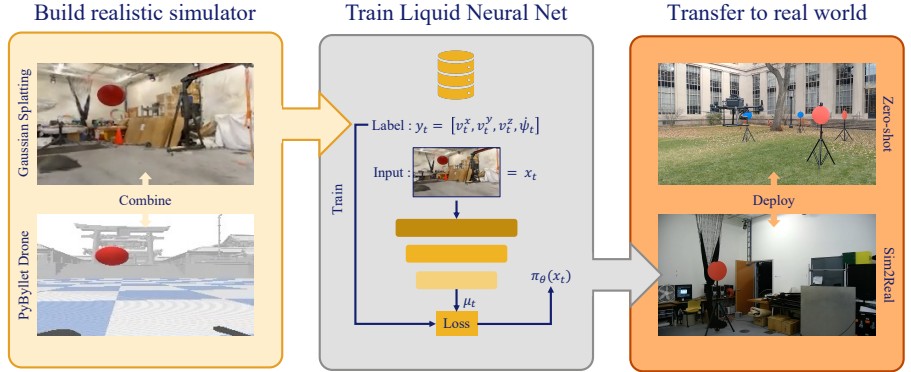

Figure 1: We create a photorealistic policy learning environment by combining Gaussian Splatting with quadrotor flight dynamics to generate high-fidelity training datasets. A model, $\pi$, with parameters $\theta$, is trained using input images $x_t$ to match its predicted output $\mu_t$ with ground truth labels $y_t = [v_t^x, v_t^y, v_t^z, \dot{\psi}_t]$, where $v_t$ denotes desired triaxial velocities and $\dot{\psi}_t$ is the yaw rate. Liquid neural networks learn the task through behavior cloning and achieve sim-to-real transfer.

8th Conference on Robot Learning (CoRL 2024), Munich, Germany.

# 1 Introduction

Recent breakthroughs in simulation now allow for the creation of more lifelike and varied datasets. Techniques like neural scene reconstruction [1, 2, 3] enable detailed, photorealistic 3D models that mirror real scenes, providing advanced training grounds for robotics. Generative AI also expands training data possibilities, with entirely new, realistic environments and scenarios created from text descriptions or similar inputs [4].

Building on these advances, we design a methodology to improve the generalization and robustness of autonomous quadrotor navigation from simulation to real-world environments. Our approach includes three schemes: 1) integrating a 3D Gaussian Splatting radiance field with a quadrotor flight dynamics engine to create a realistic simulator for generating high-quality training datasets for drone navigation; 2) designing an imitation learning pipeline to train liquid time-constant (LTC) networks [5, 6], which have out-of-distribution generalization capabilities [7], on the generated data; and 3) deploying the trained networks on a real drone to demonstrate robust generalization.

Quantitative flight tests show successful transfer of learned navigation skills from simulation to real-world settings. Policies trained exclusively on photorealistic simulated data of indoor flights targeting single manoeuvres generalize to complex, multi-step tasks in outdoor environments on real hardware. To the best of our knowledge, this is the first published report of an autonomous quadrotor policy trained entirely in simulation using imitation learning, without scene randomization, that is capable of being deployed into the real world, and generalizing to out-of-distribution environments. Our contribution relies on five key pillars:

1. **Dynamics Augmented Gaussian Splatting Simulator:** A simulator that combines the strengths of Gaussian Splatting and flight dynamics, providing both photorealistic rendering alongside high-fidelity dynamics.

2. **Implicit Closed-Loop Augmentation:** Expert trajectory design that inherently includes edge cases and recovery scenarios, eliminating the need for additional closed-loop augmentation and model fine-tuning.

3. **Liquid NNs Augmentation with Irregularly Time-Sampled Data:** Robustifying performance of Liquid NNs via irregularly time-sampled data training, and enhancing their adaptability to diverse temporal dynamics.

4. **Simulation and Real-World Validation:** Extensive testing to validate the effectiveness of our method across different simulation environments and in real-world flight scenarios.

5. **Zero-Shot Real-World Transfer:** Showcase of the pipeline's capability for zero-shot transfer, enabling a quadrotor to successfully navigate outdoor environments in a real-world hiking demonstration.

# 2 Related Works

**Imitation Learning.** Imitation learning, and more specifically behavior cloning, involves an agent learning a task by mimicking expert demonstrations, which is especially advantageous when defining a reward function is difficult. This approach captures implicit navigation knowledge in complex environments and is more sample-efficient than Reinforcement Learning (RL) [8], though it can suffer from policy stationarity and compounding errors [9]. Robot navigation methods built on foundation models [10, 11] and diffusion policies [12], exhibit very potent capabilities but violate compute constraints for real-time operability on edge devices.

The generalization problem of few-shot, one-shot, and zero-shot imitation learning agents has received extensive attention in the literature, with methods including augmentation strategies [13, 14], human interventions [15, 16], goal-conditioning [17, 18, 19, 20], reward conditioning [21, 22, 23, 24], task-embedding [25], and meta-learning [26].

We opt for this specific setting since expert drone trajectories are easy to design in simulation and thus labeled data can be produced flexibly at speed for specific flight tasks.

**Liquid Neural Networks.** Liquid Neural Networks (LNNs) are a family of continuous-time (CT) neural networks [27] that can be trained via gradient descent in modern automatic differentiation frameworks. Their building blocks are Liquid Time Constant networks (LTCs) built off of leaky-integrator neural models [28] with the steady-state dynamics of a conductance-based nonlinear synapse model [29]. The model is a differentiable dynamical system with an input-dependent varying (hence liquid) time characteristic. Outputs are obtained by numerical differential equation solvers in the ordinary differential equations formulation and by continuous functions when approximately solved as Closed-form Continuous-time (CfC) models [6]. Amongst LNNs' desirable characteristics are their stable and bounded behavior, superior expressivity [5], improved performance on time series prediction [30] and great promise in both end-to-end imitation and reinforcement learning, to map high-dimensional visual input stream of pixels to robust control decisions [7, 31, 32, 33].

Note: LNNs refer to the general category of models that are presented either by LTCs or CfCs. In this work, we use CfCs which have exhibited superior generalization performance when trained via behavior cloning for out-of-distribution flight tasks [7].

**Sim-to-Real Transfer.** Photorealistic simulators are popular for training quadrotor visual policies [34, 35, 36, 37]. Yet, photorealism is not sufficient to ensure the transfer of skills learned in simulation to the real-world, due to differences in physical properties, sensor noise, or actuator dynamics. Many sim-to-real techniques have been developed to mitigate the transfer challenge [38, 39, 40, 41]. Generalization to unseen environments is mostly achieved via domain randomization [40], which incentivizes networks to learn representations that are scene and domain-independent by randomizing input features that are superfluous to the task. Indeed, amongst existing techniques [42, 43] are position randomization, depth image noise randomization, texture randomization, size randomization, and so on.

In this work, we use a well-designed randomization in the positioning of task-relevant objects in the scene and perform RGB image augmentation on the training data (brightness, contrast, and saturation) [13, 44, 45, 46]. However, we do not resort to any randomization in the scenes simulated. In fact, we attempt to provide a simple environment data generation and training protocol and to subsequently test the fundamental task representation learning capabilities of networks.

**Gaussian Splatting.** 3D Gaussian splatting (3D GS, or GS for brevity) [2] is a technique for explicit radiance field rendering. It relies on the fitting of millions of 3D Gaussians to the training dataset images, a significantly different approach from the neural radiance field (NeRF) methodologies [1]. Offering real-time rendering without visual quality compromises, GS enables many new avenues in fields such as virtual reality and augmented reality, real-time cinematic rendering [47, 48, 49], but also autonomous driving [50]. The explicit GS construction comes with significant advantages, as it enables real-time rendering capabilities in addition to vast levels of control, compositionality and editability for complex scenario handling [51, 52].

Connections between Unmanned Aerial Vehicles (UAVs) and radiance field rendering models are well established, primarily focusing on optimizing UAV deployment for training these models. Notable methods include real-time online NeRF training from UAV camera streams [53], efficient reconstruction of unbounded large-scale scenes [54], and autonomous positioning for real-time 3D reconstruction [55], demonstrating success in various scenes and structures [56]. Additionally, GS has been utilized in Simultaneous Localization and Mapping (SLAM) for active perception modules [57] and real-time SLAM algorithms [58, 59, 60]. The simulation of quadrotor flight in neural radiance field environments is explored in [61], where planning and estimation algorithms are developed for deployment within the NeRF setting. Closest to our simulator is the PEGASUS system [3], which combines a physics engine with GS rendering and enhances objects with physical properties for manipulation data generation.

# 3 Methods

The end-to-end visual navigation model $\pi$ takes sequences of RGB images captured from drone onboard camera as input. For training, each frame $x_i$ is labeled with the instantaneous quadrotor velocities and yaw angular rate in the quadrotor body frame. More precisely, its label, $y_i$, is a 4-dimensional vector containing the x- (pitch axis, forward/backward), y- (roll axis, left/right), and z- (throttle axis, up/down) axis velocities in body-frame, in addition to the yaw rate of the quadrotor. The behavior cloning process, depicted in Fig. 1, trains the network, $\pi$, parametrized by $\theta$ to output at time $t$ a control command, $\mu_t$, given an input image, $x_t$, that minimizes a Mean Squared Error (MSE) loss function relative to the ground truth labels.

## 3.1 High-fidelity Vision and Dynamics Simulation

**GS model of the indoor flight room.** The environment selected for rendering is an indoor fight room. Images are collected using an iPhone 12 Pro camera via the Polycam app, resulting in a dataset of 430 images at $1024 \times 768$ pixels. The images are processed with COLMAP to create depth maps, and the GS model is trained on these outputs at half resolution, rendering images with satisfactory photorealism (Fig. 2) despite some visual artifacts.

**Adding models of task-related objects.** We use two objects—red and blue spheres—defined in the geometry object file format (obj). Using the open-source 3D creation suite Blender, we render images of the spheres from all angles. We then apply the same training procedure as before to train a Gaussian Splatting model for each object. Finally, we manually tune the size and intensity of the trained GS models to ensure smooth blending into the indoor environment, as shown in Fig. 2.

**Dynamics for GS trajectories.** To generate images reflecting the dynamics of a simulated quadrotor, we align the visual GS world with the dynamics solver by finding the gravity-aligned vertical vector. Using the PyBullet physics engine [62], we tune the scaling to ensure consistent object positioning between the environments. The physics simulation runs at 240Hz, while inference occurs at a slower rate. The GS camera moves based on relative displacement in PyBullet, rendering updated views from the quadrotor's positions for trajectory data or neural network inference (see Fig. 3). Implementation details are in Appendix A.

## 3.2 Task Description

The task is to navigate through multiple checkpoints using visual cues. The agent must reach the current target and turn according to its color: left for red and right for blue, as shown in Fig. 2. This setup mimics following trail blazes on a hike, guiding the hiker both in direction and toward the next cue. Consequently, each expert demonstration trajectory consists of two phases: approach and turn. We generate an equal number of right and left trajectories to avoid bias in the models.

**Approach.** Naive Imitation Learning for autonomous navigation often fails when encountering trajectories outside the training distribution. To address this, recovery trajectories—showing how to return to desired paths from suboptimal positions—are added to the training data. This method is usually infeasible with real-world demonstrations but can be synthesized. Here, we ensure all trajectories contain recovery information by positioning the quadrotor at trajectory starts so the target partially appears at the image border and by randomizing the initial target distance. This exposes the network to various approach angles and distances, driven by PID controllers that adjust the quadrotor's velocities and yaw rate to center the target in the frame. The significance of this strategy is discussed in Appendix C.4.

**Turn.** The turn procedure is straightforward and requires no special treatment, as it begins once the drone hovers at a fixed distance from the target. We send a constant yaw rate command, with an amplitude of about $12°$ per second—nearly double the maximum rate during the approach phase when the sphere is at the frame's edge. This clearly distinguishes the switch in behavior at test time, reducing the risk of confusing a segment of the approach maneuver with a decision to turn.

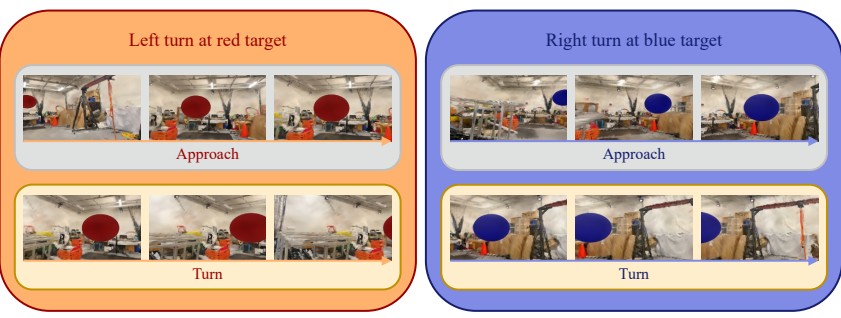

Figure 2: Gaussian Splatting dataset sample trajectories. The trajectories are designed in two phases, first an approach where the drone gets close to the target and centers it in the middle of the image, followed by a turn procedure in the direction corresponding to the color of the target.

### 3.3 Irregularly Sampled Data Augmentation

It is often assumed that the inference frequency of a neural network at test time is that of the training data. This holds when data collection and model deployment are both executed on the same platform and data processing and inference times are negligible with respect to the sample period. In fact, popular recurrent models cannot handle data that is irregularly sampled or sampled at a different frequency, as they treat time steps between inputs as a constant during training.

Neural ODEs [63], however, can be evaluated at arbitrary time intervals, as their hidden states $\mathbf{x}(t)$ evolve according to the time-continuous differential equation, $\frac{d\mathbf{x}(t)}{dt} = f(\mathbf{x}(t), \mathbf{I}(t), t, \theta)$.

More specifically, the closed-form CfC representation of the hidden states is given by [6]:

$$\mathbf{x}(t) = \sigma(-f(\mathbf{x}, \mathbf{I}; \theta_f)t) \odot g(\mathbf{x}, \mathbf{I}; \theta_g) + \big[1 - \sigma(-[f(\mathbf{x}, \mathbf{I}; \theta_f)]t)\big] \odot h(\mathbf{x}, \mathbf{I}; \theta_h) \qquad (1)$$

Here, $\mathbf{x}(t)$ is the hidden state, $\odot$ is the Hadamard product, $\sigma$ is the sigmoid function, $f$, $g$, and $h$ are three neural network heads with a shared backbone, parameterized by $\theta_f$, $\theta_g$, and $\theta_h$, respectively. $\mathbf{I}(t)$ is an external input, and $t$ is time sampled by input time-stamps, as illustrated in Fig. 10 in Appendix D.

As time appears explicitly in equation (1), the formulation eliminates the need for complex numerical solvers (speedup between one and five orders of magnitude). We utilize this flexible data handling to implement an irregularly sampled data augmentation scheme that randomizes the time steps between training samples, helping networks generalize across frequency domains.

## 4 Experiments

### 4.1 Hardware Platform

All real-world evaluations of trained policies are executed on a DJI M300 RTK quadcopter. The M300 interfaces with the DJI Manifold 2 companion computer, enabling programmatic control of the drone. The onboard computer runs an NVIDIA Jetson TX2, which has GPU capability for onboard inference in real-time (more information about the hardware setup in Appendix A).

### 4.2 Models Deployed

All models share the same general architecture of a Convolutional Neural Network (CNN) backbone that extracts features from the input images ($256 \times 144$ pixels) and a Recurrent Neural Network (RNN) head for sequential decision-making, trained with the MSE loss. Details about the training procedure and model architectures are provided in Appendix D.

**Liquid variants.** We train three CfC models to test the impact of training data frequency and time scale augmentation on performance. All three models share the same skeleton of 34 liquid neurons,

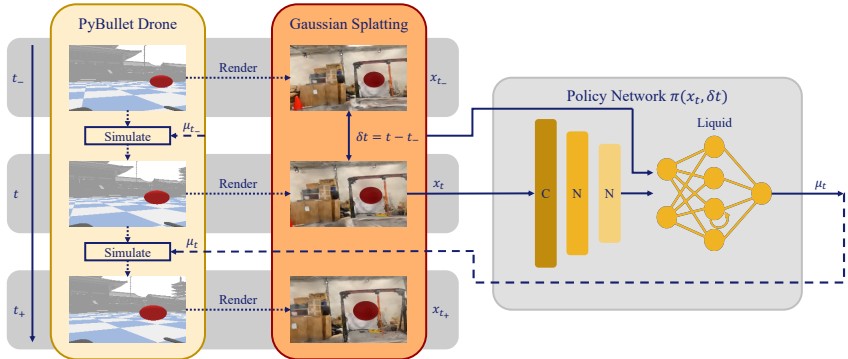

Figure 3: Diagram of the in-simulation inference protocol with physics engine dynamics and GS rendering. At time $t$, the GS image $x_t$ is rendered from the simulated position and attitude, and processed through the CNN of the policy network $\pi(x_t)$. Obtained features, along with the elapsed time $\delta t = t - t_-$, are input into the Liquid network, which outputs the velocity control command.

and are trained on the same collection of trajectories. However, the frequencies at which the data points are logged along the trajectories are modulated to generate three different datasets.

The first model, called Liquid, is designed for time-scale augmentation. The frequency of recorded images and labels varies throughout the trajectories, with each data point taken at a random time difference corresponding to frequencies between 1 and 10 Hz (see Appendix B). The dataset includes input images, labels, and the time delta since the last input, as discussed in section 3.3. Liquid 3Hz and Liquid 9Hz are trained on datasets with constant sampling rates of 3Hz and 9Hz, respectively.

**LSTM Baseline Model.** Long Short-term Memory (LSTM) [64] is a discrete-time neural network that consists of a memory cell that is controlled by learnable gates that access, store, clear, and retrieve the information contained in the memory. Amongst a plethora of RNN architectures tested on similar visual flight navigation tests, the LSTM architecture seems to offer the best performance amid the non-liquid neural networks [7]. Thus, we select it as our baseline and train on the 3Hz dataset to maintain proximity to the test hardware running frequency (LSTM cannot handle irregularly sampled data). The LSTM architecture we devise comprises just over 300k parameters. It is worth noting that its size is about 25 times that of its liquid counterparts that we deploy in this work.

### 4.3 Results

**Evaluation protocol.** Performance is associated to manoeuvre success rates. For a checkpoint to be valid, we require the quadrotor to come to a hover at a distance corresponding to the stopping position seen in training (0.5 m in simulation, between 1 and 1.5 m for the real platform) and then turn in the correct direction at a rate close to that used of the labels (above $10°$ per second). Failure cases are defined in Appendix C.

**Sim-to-Sim.** The initial setup consists of training with the non-photorealistic input images from the PyBullet simulator. This enables longer and more elaborate hiking tests, exploiting the flexibility and unlimited space available. We evaluate the models at an inference frequency of 3 Hz on a 100-step hiking task, with randomly initialized targets ensuring the next checkpoint is centered in the drone's field of view after correct turns. The bar plot showing the lengths of the hikes completed by each agent for 10 different initial checkpoint placements is provided in Fig 4.

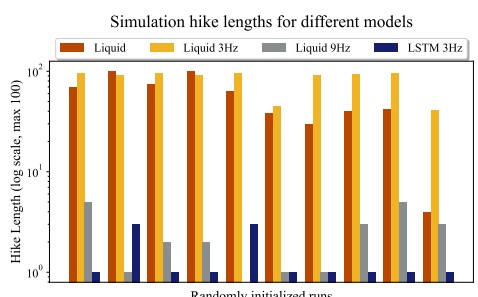

Figure 4: Results of the infinite simulated hike experiment for 10 different initializations of the goal positions in the PyBullet environment.

Liquid and Liquid 3Hz models consistently achieve long hikes (mean lengths of 56.1 and 83.4, respectively), successfully performing both the 1-target task and chronological composition. The Liquid 9Hz agent averages only 2.3 checkpoints, sometimes failing the first target manoeuvre, highlighting challenges in closed-loop inference at different frequencies from training. Elsewhere, the model really struggled to perform the task on blue targets. The LSTM model can perform the 1-target task consistently but fails to generalize to multiple steps as it generates more erratic commands (1.4 hike length on average).

**GS-to-GS.** The models trained on the GS datasets are evaluated in the GS-rendered simulator at an inference frequency of 3 Hz, following the process shown in Fig. 3. Space constraints in the rendered indoor flight room limit the number of targets that can reasonably fit into the scene to two. Agents have a limited time window to navigate the two checkpoints, with success rates for the first step based on color and the average steps completed shown in Table 1.

| Model | $1^{st}$ target rate | | | 2 target |
| | Red | Blue | Total | Mean |
| --- | --- | --- | --- | --- |
| Liquid | **0.94** | **0.98** | **0.96** | **1.72** |
| Liquid 3Hz | 0.64 | **0.96** | 0.79 | 1.35 |
| Liquid 9Hz | 0.57 | 0.72 | 0.64 | 1.00 |
| LSTM | **0.98** | 0.43 | 0.72 | 1.04 |

Table 1: GS-to-GS evaluation results of models on 2-target runs in the GS indoor flight room scene ($N = 100$ with color decided by a coin flip, 53 runs start with a red target and 47 with blue).

The first observation is the Liquid 9Hz agent's incapacity to compete with the other Liquid models, highlighting how training data frequency impacts performance at different rates. Amongst the other models, we notice that both LSTM and Liquid 3Hz exhibit a bias in performance in favor of one of the colors, respectively red and blue (both over 96% success rate), doing much worse on the other. With only 43% on blue, LSTM achieves an average of 1.04 out of 2 hike steps. Similarly, but to a lesser extent, Liquid 3Hz struggles with red checkpoints (64% success rate), reaching an average length of 1.35 checkpoints. The Liquid agent seems to have equally well-mastered left and right manoeuvres (94 and 98% success respectively) and completes longer hikes, 1.72 steps on average. It also exhibits the most efficient and smooth manoeuvring (details in Appendix C).

**GS-to-Real.** The sim-to-real transfer evaluation, in the indoor flight room used for training the GS rendering, is deployed on the Matrice 300 quadrotor platform. We use cardboard disks as the goals. In turn, we position the targets on a tripod in the center of the room at a height of about 2 meters. We select five starting positions around the target, placing the quadrotor 4 to 5 meters away to give each agent four attempts at each color and expose them to multiple room angles. The translation velocities inferred by the networks are scaled in such a way as to balance enough time for human intervention and repeated test efficiency. The per color and total success rates are given in Table 2.

| Model | Success rate | | |
| | Red | Blue | Total |
| --- | --- | --- | --- |
| Liquid | **0.8** | 0.7 | **0.75** |
| Liquid 3Hz | 0.25 | **0.85** | 0.55 |
| Liquid 9Hz | 0.5 | 0.6 | 0.55 |
| LSTM | 0.05 | 0.2 | 0.125 |

Table 2: GS-to-Real evaluation results of models on single target runs in the indoor flight room ($N = 40$).

The challenge of direct sim-to-real Transfer is epitomized by the large drop in performance suffered by the LSTM agent (from 0.72 in GS to 0.125). Liquid neural networks, on the other hand, suffer minimally in comparison. Indeed, both the Liquid 3Hz and 9Hz agents succeed just over half of the time. Again, it is the time-scale augmented Liquid agent that performs best. It goes from 96 to 75% success rate, maintaining balanced performance across colors.

**GS-to-Real + Generalization.** We deploy the same test protocol on an outdoor lawn in an urban campus. In addition to the completely unseen scene, the various starting positions now expose the agents to sunlight from various angles, making for a challenging generalization scenario. We compare the best Liquid model and the LSTM model for reference in this setting, and the results are provided in Table 3.

Unsurprisingly, the LSTM agent is incapable of completing the task even once in 40 attempts. Our Liquid agent, however, demonstrates zero-shot task completion capability. Though relatively struggling with red targets (40%) in the outdoors setting it does better on blue checkpoints (60%), exhibiting the ability to perform the simulation-learned task more than half of the time completely out-of-distribution outdoor real-world setting. Challenging background lighting conditions, as well as tree shades and reflections off buildings impact the appearance and robust perception of targets.

Encouraged by these results, we design a multi-step outdoor setup in an alternative campus location to showcase performance on the comprehensive hiking task. Our Liquid agent demonstrates state-of-the-art transfer and generalization by successfully navigating through a series of 5 consecutive checkpoints. Videos of training data runs, experiments and real world demos are provided on the project website (link).

| Model | Success rate | | |
|-------|-----|------|-------|
|  | Red | Blue | Total |
| Liquid | **0.4** | **0.6** | **0.5** |
| LSTM | 0 | 0 | 0 |

Table 3: GS-to-Real zero-shot evaluation results of models on single target runs on the outdoor campus lawn ($N = 40$).

## 5 Limitations

**Dynamic scenes:** Gaussian Splatting, like other neural rendering techniques, excels in static scenes but struggles with dynamic environments, such as moving objects and changing conditions, limiting its effectiveness in generating dynamic data for real-world scenarios. While affine transformations can adjust Gaussian attributes for moving objects, this process increases rendering time and is constrained to discrete components. In contrast, Liquid neural networks demonstrate robustness to dynamic backgrounds during testing [7], allowing them to focus on tasks despite moving distractions. However, efficiently learning from dynamic data in applications like urban air mobility and wildlife tracking remains outside the scope of this work.

**Trajectory augmentation:** As tasks become more complex and require longer planning horizons, designing efficient trajectory augmentation schemes becomes increasingly challenging. Current methods may not cover the full range of scenarios, especially edge cases and recovery situations, which are crucial for robust performance. This general limitation of imitation learning holds for datasets collected in the real world, in simulation, and in rendered environments.

**Distribution shifts:** Large distribution shifts between training and real-world environments can severely hinder performance; thus, the need for domain randomization to bridge substantial gaps remains essential for robustness across real-world environments not encountered during training.

Addressing these limitations is crucial for improving the robustness and applicability of visual end-to-end autonomous navigation systems trained in simulation for diverse real-world scenarios.

## 6 Conclusion

3D Gaussian Splatting has emerged as a potential solution for generating realistic visual training data flexibly and at scale for autonomous systems, surpassing the limitations of real-world data collection. However, bridging the gap between these simulated training environments and real-world applications remains a significant challenge in robot learning, typically addressed through domain randomization. In this work, we propose an end-to-end protocol for training autonomous agents for deployment into the real world, encompassing data-driven simulation methods, and reliance on Liquid neural networks, with newly explored augmentation techniques for improved transfer and generalization. Comprehensive experiments empirically confirm the ability of learned agents obtained via our pipeline to directly be deployed on an autonomous quadrotor platform to successfully perform a hiking task in environments they have never seen before. We believe our approach represents a leap in the employment of Gaussian Splatting and Liquid neural networks for training robust visual end-to-end autonomous navigation policies for direct, real-world deployment.

**Acknowledgments**

Support for this research has been provided in part by the Boeing company and ONR's Science of Autonomy program. We are grateful for it.

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

# Appendices

## A    Simulator and hardware setup

### A.1    Indoor room GS model

Images of the room are collected using a consumer smartphone (iPhone 12 Pro) camera through the free access application Polycam. The application automates the capture of frames, recording images when the novelty in the frame is sufficiently large. We thus obtain a dataset of 430 images of the room at a resolution of $1024 \times 768$ pixels. The images are processed with the general-purpose Structure-from-Motion (SfM) pipeline COLMAP [65] to create depth maps. The GS model is trained on the COLMAP outputs at half resolution ($512 \times 384$), densification interval of 500, and 30,000 iterations. Training takes around 15 minutes on a single GPU (NVIDIA GeForce RTX 3080 Ti). The obtained model renders images of satisfactory photorealism as seen in Fig. 2, however, with the presence of many objects in the room and limited training, there remains some fogginess and some visual artifacts in the images. The purpose of this work is to achieve robust generalization, rather than to improve radiance field rendering. Thus we did not pursue further improvements of the GS model.

### A.2    Grounding GS dynamics

We ensure that the points of view we use for the generation of images are on trajectories true to the dynamics of an accurately simulated quadrotor model. To do so, we first use the GS renderer user interface to approximately find the gravity-aligned vertical vector. This is necessary to align reference frames between the visual GS world and the dynamics solver. PyBullet physics is used in our simulator which we adapt from the codebase of the authors in [62]. The drone dynamics we use are based on Bitcraze's Crazyflie 2.x nano-quadrotor.

Scaling is tuned manually such that distance units in PyBullet and units in the corresponding GS environment match. Furthermore, we ensure that the positioning of objects matches: for a given position and attitude, the drone front image extracted from the PyBullet environment and that obtained from the GS rendering contains the object of the same size and in the same position (as depicted in Fig. 3). The simulator thus behaves as a direct photorealistic enhancement of images seen from trajectories in the PyBullet physics simulator. Hence, each trajectory can be visualized in both environments and datasets can be generated for either.

### A.3    Simulation protocol

The physics simulation runs at 240Hz, and we allow low-level flight control to occur at the same frequency, although inference runs at a much slower rate. Indeed, for inference or data collection, we simulate the evolution of the system with constant velocity command for a number of steps corresponding to the desired period. Then, we compute the relative displacement in the PyBullet environment and convert it to the GS space's dimensions. Next, we move the GS camera accordingly and render the view from the updated quadrotor position and attitude. Finally, the captured image is added to the trajectory data or fed into the neural network for inference before repeating with the current velocity command (process visualized in Fig. 3).

### A.4    Hardware details

The DJI Onboard SDK (Software Development Kit) and its associated ROS wrapper provide an interface for feeding the drone's low-level flight controller with desired high-level translation velocities and yaw rate commands. The flight controller itself is a black box system provided by the manufacturer which controls the four-rotor speeds to track the velocities specified by the TX2 computer. It is worth mentioning that the dynamics of the platform we use do not match those of the

nano-quadrotors simulated. A Zenmuse Z30 camera and gimbal are mounted on the underside of the drone, pointed forward. The gimbal compensates for the drone's current orientation, such that the roll or pitch of the drone does not result in the roll or pitch of the camera image. The gimbal follows the drone's current yaw such that the camera is always pointed forward. Input images gathered by the gimbal-stabilized camera are available to the companion computer via the SDK. We reach a runtime frequency of around 2.6 Hz with our setup, a budget shared between image capture, communication with the onboard computer, pre-processing and model inference, and sending of the output command.

# B Dataset design

## B.1 Trajectory design supplemental details

Designing our dataset, we limit the trajectories to the case of reaching and turning at a single target, as opposed to generating runs hopping through multiple checkpoints. This choice is deliberate as it gives us more control over the initialization augmentation procedure, leading to better training data efficiency while ensuring the multi-step capability of the networks through sequential composition.

In the simulator, we always assume that the target is positioned at the origin of the $(x, y)$ horizontal plane, and at a fixed altitude. Each new trajectory is generated using PID controllers that stabilize the yaw, altitude, and distance gaps to a centered position facing the target. We uniformly sample an angular position anywhere around the origin, a starting distance from the target in a prefixed range. We then randomize the initial attitude and altitude of the drone as to randomize the placement of the target in the first frame. Three quarters of trajectories are such that the target is at a lateral edge (vertical position uniformly sampled). In the remaining runs, the target is at a vertical edge, with uniformly distributed yaw. This procedure is summarized in Algorithm 1 and a visualization of the positions of the spheres in the first frame of a 100 sampled trajectories is provided in Fig. 5.

---

**Algorithm 1 : Initial position and attitude sampling**

**Require:** Maximum yaw angle $\psi_M$, Distance range $[d_m, d_M]$, Altitude range $[z_m, z_M]$

```
# Initialize roll and pitch to zero:
```
$(\theta_0, \phi_0) = (0, 0)$
```
# Sample initial angle and distance:
```
$\theta \sim \mathcal{U}(0, 2\pi) \, , d \sim \mathcal{U}(d_m, d_M)$
```
# Position the drone in the horizontal plane:
```
$(x_0, y_0) = (d \, cos(\theta), d \, sin(\theta))$
```
# Sample chance parameter:
```
$c \sim \mathcal{U}(0, 1)$
```
# Select yaw and altitude accordingly:
```
**if** $c < 0.75$ **then**
  $z_0 \sim \mathcal{U}(z_m, z_M)$
  $\psi_0 \sim \mathcal{B}(0.5, \{-\psi_M, +\psi_M\})$
**else**
  $z_0 \sim \mathcal{B}(0.5, \{z_m, z_M\})$
  $\psi_0 \sim \mathcal{U}(-\psi_M, +\psi_M)$
**end if**
**return** $s_0 = [x_0, y_0, x_0, \theta_0, \phi_0, \psi_0]$

---

We also provide sample trajectories for 600 approach phases generated according to the above described protocol with the ground-truth informed PID expert controller in Figure 6 and Figure 7. The trajectories show a dense coverage of the entire physical space available between the drone and the target in terms of both lateral and vertical displacements, thus including recovery information from a large set of situations by design.

## B.2 $\delta t$ probability mass function

The time deltas are sampled to be divisible at the resolution of our simulation frequency (240Hz). Time delta in seconds, $\delta t$, is sampled by the following distribution:

$$\delta t \sim \delta t_{min} \left[ \alpha \, \mathcal{N}(\mu_1, \sigma_1) + \beta \, \mathcal{N}(\mu_2, \sigma_2) \right] \tag{2}$$

with $\delta t_{min}$ being the physics simulation unit time step, $\mathcal{N}(\mu, \sigma)$ a Gaussian distribution of mean $\mu$ and standard deviation $\sigma$, and $\alpha$, $\beta$, $\mu_1$, $\sigma_1$, $\mu_2$, and $\sigma_2$ constants tuned to ensure adequate training run lengths (we require a minimum of 64 frames for a single trajectory), while still representing the full range of values between 1 and 10 Hz. The probability mass function can be seen in Figure 8.

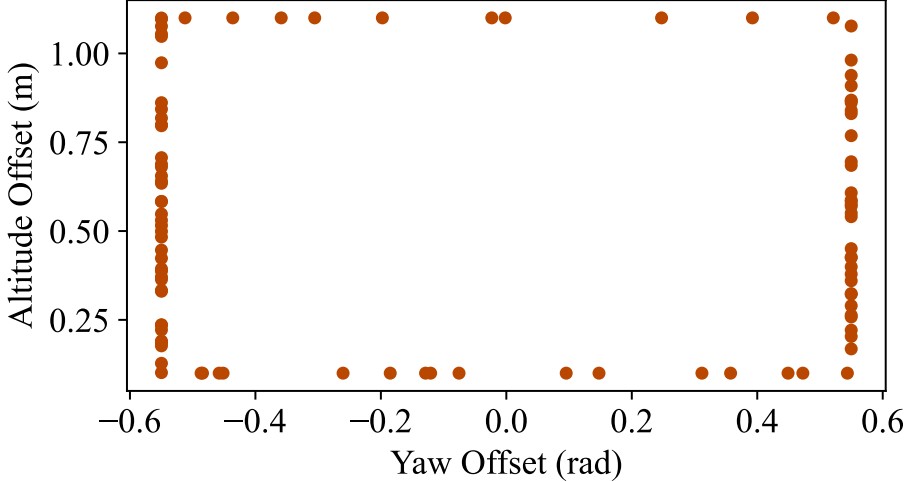

Figure 5: Distribution of 100 initial altitudes and yaw offsets for trajectory generation.

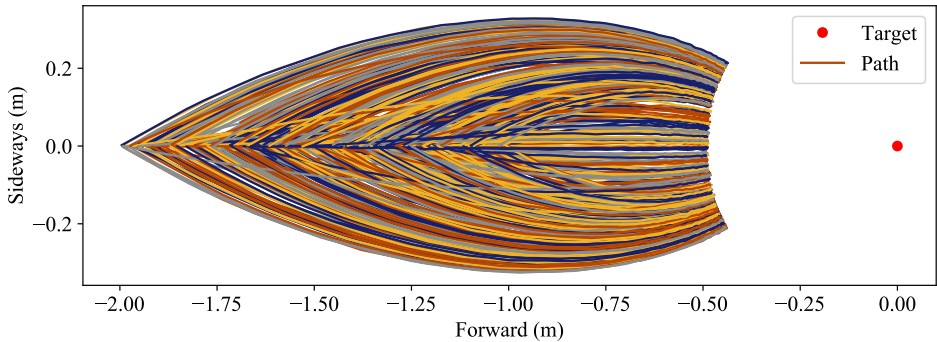

Figure 6: Lateral plane projection of training dataset trajectories. Trajectories are curved as both yaw and forward movements are actuated simultaneously (n=600).

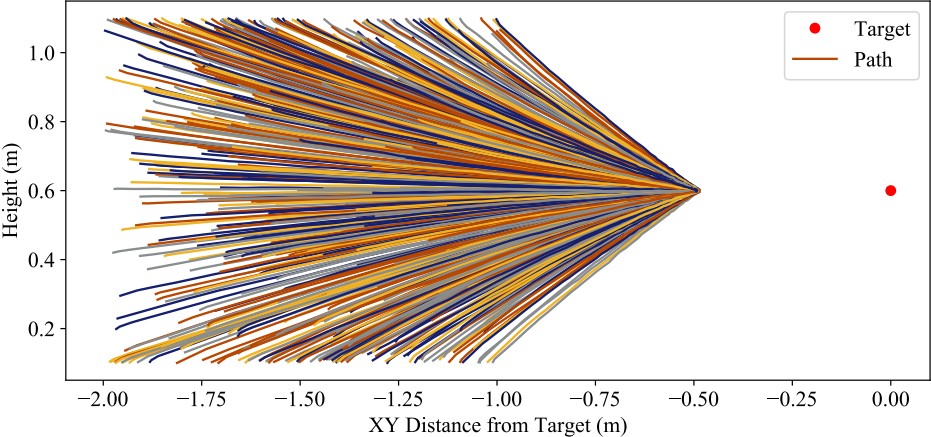

Figure 7: Vertical plane projection of training dataset trajectories. Trajectories depict the quadrotor rising or diving to align with the target (n=600).

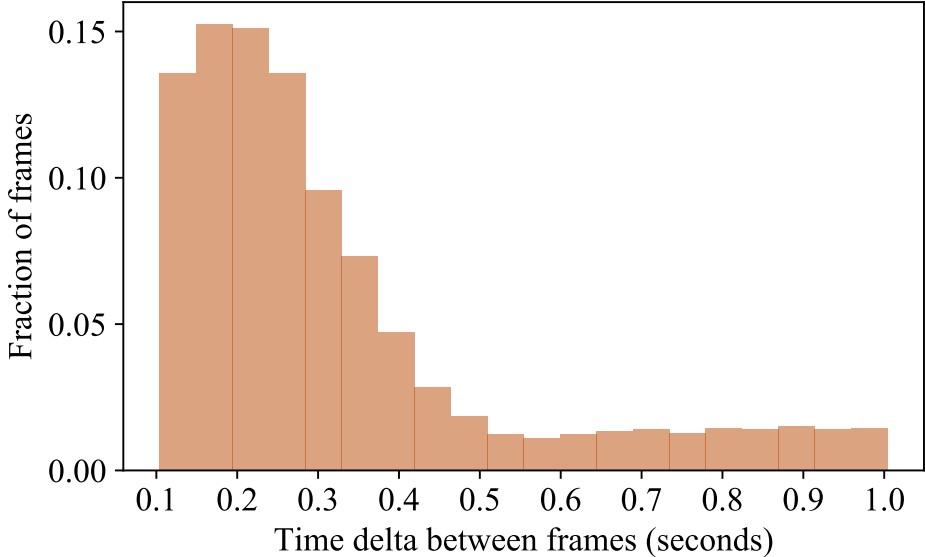

Figure 8: Probability mass function of the $\delta t$ sampled for the generation of irregularly sampled datasets with the tuned parameters $\alpha = 3$, $\beta = 1$, $\mu_1 = 21$, $\sigma_1 = 30$, $\mu_2 = 196$, $\sigma_2 = 100$, and $t_{min} = \frac{1}{240}$.

# C   Additional Results

## C.1   Failure case identification

Situations where the drone shoots past the targets (no slowing down), turns in the wrong direction (turn triggered at the rates of interest but with the opposite sign), stays still for prolonged periods (sub centimeter per second forward velocity during the approach phase for over 20 seconds in the real world tests, or if taking more than twice the average time of a manoeuvre in simulation), or loses the target before completing the approach (target out of frame during the approach and/or approaching another visual cue), all constitute failure cases. In our experiments, we notice that the most common mode of failure is the loss of the target out of the field of view during the approach phase (Sim, GS and Real) as well as turns in the wrong direction after a successful approach (GS and Real). Cases where the agents get stuck indefinitely occurred a handful of times across testing platforms.

## C.2   Sim-to-GS

The successful transfer of task completion capability from the GS trained liquid models to the real world poses the question of the level of photo-realism required to sustain such a transfer. Thus, we test whether models trained with non-rendered simulation data can generalize to the GS based simulation evaluation. We see this Sim-to-GS as an intermediate step between Sim-to-Sim and Sim-to-Real, which would most likely be informative about the prospects of achieving the latter. The results presented in Table 4 shed light on the difficulty agents have in generalizing meaningful behavior to environments with drastic visual characteristic changes. In fact, the Liquid 3Hz and 9Hz models suffer from a massive color-blind biases towards right turns and LSTM towards the left. The Liquid agent fails to initialize turns which leads to an extremely low performance. Thus we verify our claim that photo-realism is crucial for transferring vision-only end-to-end tasks learned in simulation.

Table 4: Sim-to-GS evaluation results of models on 2-target runs in the GS indoor flight room scene ($N = 100$ with color decided by a coin flip. 53 runs start with a red target and 47 with blue).

| Model | 1st-target success rate | | | 2-target hike length |
|---|---|---|---|---|
| | Red | Blue | Total | Mean |
| Liquid | 0.04 | 0.02 | 0.03 | 0.015 |
| Liquid 3Hz | 0.13 | 0.04 | 0.09 | 0.05 |
| Liquid 9Hz | 0.43 | 0.00 | 0.23 | 0.15 |
| LSTM | 0.00 | 0.13 | 0.06 | 0.03 |

## C.3   Half duration GS-to-GS

Elsewhere, we rerun the GS-to-GS experiment after halving the time budget allocated to the agents to complete the 2-step hike. This allows us to evaluate the speed and efficiency closed-loop navigation exhibited by the different networks. The results, provided in Table 5, show the superiority of the Liquid agent which performs at the same standards in half the time (from 1.72 to 1.66 on 2-target score), while the LSTM performance is halved on the 2-target metric (1.04 to 0.51) and drops significantly for the first target as well (0.72 to 0.51). Both regularly time sampled Liquid models fail to complete the second passage in the reduced time (apart from 2 full runs completed by Liquid 3Hz).

## C.4   Importance of recovery trajectories

We compare 5 trained instances of the Liquid model with the data generation scheme described in the paper (Full Window), 5 instances on a dataset generated with the same procedure but with the window (along which initial positions are sampled) halved in height and width (Half Window), and 5 instances on a dataset generated by randomly sampling the original yaw and height as to have the

Table 5: Half time budget GS-to-GS evaluation results of models on 2-target runs in the GS indoor flight room scene ($N = 100$ with color decided by a coin flip. 53 runs start with a red target and 47 with blue).

| Model | 1st-target success rate | | | 2-target hike length |
|---|---|---|---|---|
| | Red | Blue | Total | Mean |
| Liquid | **0.96** | **1.0** | **0.98** | **1.66** |
| Liquid 3Hz | 0.64 | 0.89 | 0.76 | 0.79 |
| Liquid 9Hz | 0.57 | 0.72 | 0.64 | 0.64 |
| LSTM | 0.77 | 0.21 | 0.51 | 0.51 |

target uniformly dispersed in the initial frame (Uniform Sampling). These are done in the PyBullet simulation environment and the best checkpoint based on validation error is saved for each.

All 15 models are first tested on the Sim-to-Sim experiment as reported in the manuscript's Section 4.3. The average hike lengths for each training iteration of each model (across the same 10 initializations used in 4.3) are provided in Table 6.

Table 6: Average hike length for different recovery strategies

| Training Instance | Full Window | Half Window | Uniform Sampling |
|---|---|---|---|
| 1 | 94.4 | 11.1 | 5.2 |
| 2 | 4.5 | 9.5 | 10.3 |
| 3 | 5.1 | 0.0 | 83.0 |
| 4 | 94.2 | 2.5 | 58.2 |
| 5 | 7.2 | 2.6 | 88.9 |
| **Average** | 41.1 | 5.1 | 49.1 |

It is clear that sampling initial positions that never reach the sides and corners of the initial image is not a good strategy (even here, where testing does not involve extreme recoveries). Two training instances are around the 10 average hike length with the Half Window recovery dataset, with the others considerably worse. Our Full Window recovery strategy generates the best-performing models overall, although other training instances lead to poor performance. Uniform Sampling produces slightly less high-scoring models as well, with an improved average due to a smaller drop-off for lesser runs.

We then set up a recovery-specific test, where we force the initial placement of the target in one of the four corners of the frame, and run 100 single-target reach and turn tests with a fixed time window for each training iteration as above (discarding the disqualified Half Window strategy). The success rates are given in Table 7.

Table 7: Recovery importance target-in-corner test results comparing Full Window and Uniform Sampling. Values are success rates ($N = 100$).

| Training Instance | Full Window Recovery | Uniform Sampling |
|---|---|---|
| 1 | 1.00 | 0.75 |
| 2 | 0.73 | 0.83 |
| 3 | 0.89 | 0.60 |
| 4 | 1.00 | 0.64 |
| 5 | 0.84 | 0.74 |
| **Average** | 0.89 | 0.71 |

We notice that while uniform sampling can only lead to a single model performing above the 80% mark, our full window approach only leads to a single checkpoint that does not reach that figure with all others well above and two out of five with perfect performance. Overall, this leads to an

18% improvement in performance on average and highlights the importance of maximizing recovery trajectories in the training dataset to ensure robustness.

## C.5    Effect of irregularly sampled data augmentation on LSTM

To assert the architectural superiority of liquid neural networks we modify the LSTM architecture to include the irregularly sampled time difference to the last frame as an additional input. We call this variant LSTM+$\delta t$. We train 5 instances of the LSTM+$\delta t$ (with the LSTM hyperparameters) to compare to 5 retrained Liquid models on the PyBullet simulator data (same checkpoints as above). The models are all tested on the Sim-to-Sim 100 step hike. The average hike length obtained among the 10 initializations used in Section 4.3, as well as their average across training instances, are provided in Table 8.

Table 8: Average hike length for Liquid vs. LSTM+$\delta t$

| Training Instance | Liquid | LSTM+$\delta$t |
|---:|:---:|:---:|
| 1 | 94.4 | 9.9 |
| 2 | 4.5 | 9.8 |
| 3 | 5.1 | 0.6 |
| 4 | 94.2 | 3.3 |
| 5 | 7.2 | 6.2 |
| **Average** | 41.1 | 6.0 |

The best LSTM+$\delta t$ model obtained can barely reach 10 targets per run, which is under a fifth of the score reported in Section 4.3 for Liquid (56.1) and close to ten times less than the best-obtained model after retraining (94.4). It does, however, perform better than the LSTM model reported in the paper (average of 1.4 hike length). This shows that augmenting models with time-step knowledge is indeed beneficial, but much larger performance discrepancies are to be attributed to model architecture.

## C.6    Real-world trajectory visualisation

Figure 9 provides a top-down view of the drone's trajectory in the horizontal plane during the demonstration provided in our supplementary material (video with 5 checkpoints in the completely out-of-distribution environment). The blue and red dots represent the approximate reconstructed locations of the targets along the hike. We use a colorbar to visualize the variations in forward velocity $v_x$. The trajectory starts in the bottom left corner and the drone makes its way up to the top right. We notice that as programmed in the training data, the forward velocity is relatively high during approach and then reduces to a near stop before turning in front of the targets. Although robust in navigating in a similar fashion to the proportional gain controller from training (increasingly aligned with target, no systematic overshoot), the smoothness and duration of transitions are somewhat variable due to control and time delay errors, exterior conditions (wind), and the high level of distribution shift that the model has to deal with.

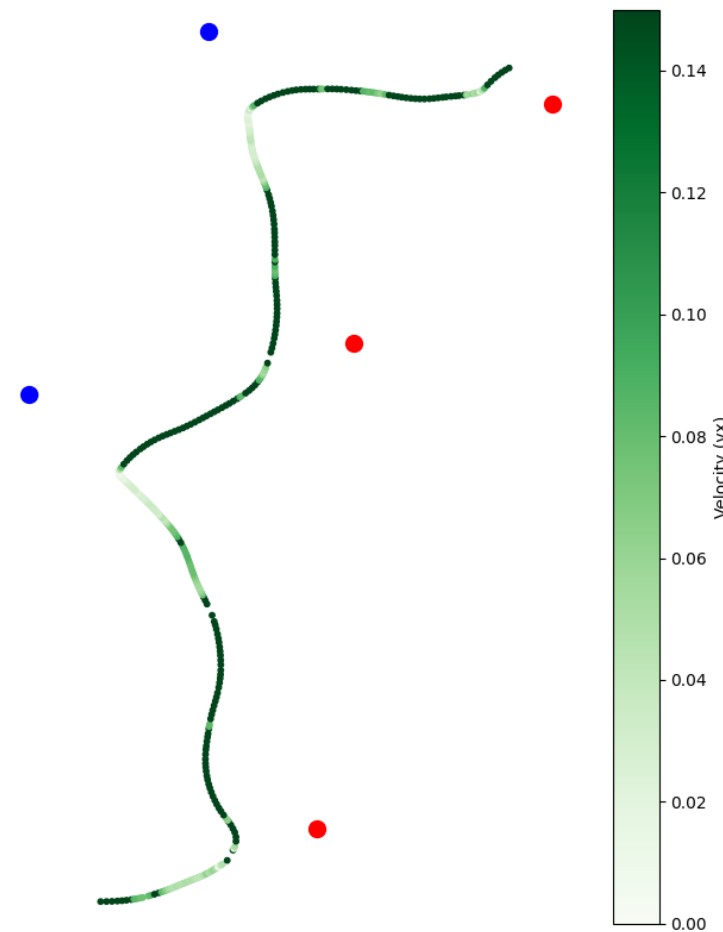

Figure 9: Top-down visualisation of real-world five step hike in an out of distribution environment. The quadrotor flight starts in the bottom left and ends in the top right corner of the figure.

# D  Training and models

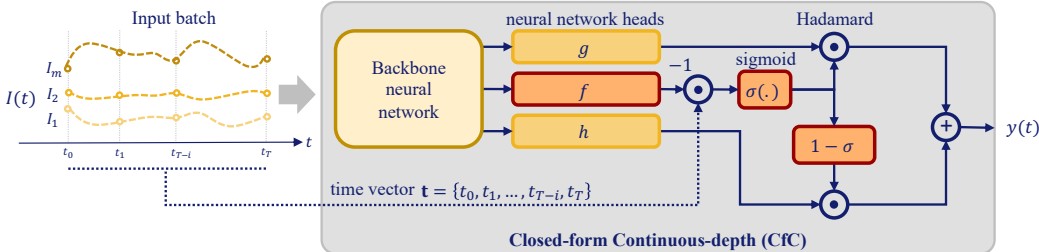

Figure 10: Liquid (Closed-form Continuous-depth) neural architecture. A backbone neural network layer delivers the input signals into three head networks $g$, $f$, and $h$. $f$ acts as a liquid time constant for the sigmoidal time-gates of the network. $g$ and $h$ construct the nonlinearities of the overall CfC network. (Adapted from [6] with the authors' permission.)

## D.1  Training and model parameters

We balance the number of training epochs and the number of trajectories with the different frequencies by ensuring all models are trained with roughly the same number of training batches. We use 200 as the minimum number of trajectories for the 9Hz model (100 red and 100 blue). We use 600 training trajectories for the 3Hz models to use roughly the same number of frames. However, since there is a nonlinear relationship between the number of batches and the length of a single trajectory, we train for twice as many epochs in the 3Hz models to get roughly the same total number of training batches within a training run.

We use Tree-structured Parzen estimators (TPE) [66, 67], a sequential model-based optimization algorithm, to tune a subset of hyperparameters for each of the models (some values are hand-selected to reduce the search space dimension). The Optuna library [68] handles the hyperparameter optimization, exerting early pruning of models with a validation loss higher than the median of previously tested models ($n > 3$) after 10 epochs. Tuning includes the learning rate (LR), LR decay rate, and model-specific hyperparameters such as backbone unit count and connection seed. The combination of hyperparameters achieving the smallest validation loss over 40 trials of 50 epochs is used for final training. Each architecture is trained 5 times with the optimized hyperparameters, and the best-performing checkpoint on the validation loss metric is then deployed for testing. Model and training procedure parameters can be found in Table 9 and Table 10.

Table 9: **Model hyperparameters:** Best hyperparameters for each model found via TPE sampling. Fixed parameters were set ahead of time. (n/a = not applicable)

| Param Name | Liquid | Liquid 3Hz | Liquid 9Hz | LSTM |
|---|---|---|---|---|
| RNN size | 98 | 133 | 145 | 220 |
| Learning rate ($\times 10^{-4}$) | 4.41 | 5.95 | 13.69 | 0.33 |
| LR decay rate | 0.87 | 0.95 | 0.99 | 0.99 |
| Dropout | n/a | n/a | n/a | 0.016 |
| Fixed (not tuned) parameters | | | | |
| Connection seed | 22224 | 22224 | 22224 | n/a |
| Inter neurons | 18 | 18 | 18 | n/a |
| Command neurons | 12 | 12 | 12 | n/a |
| Motor neurons | 4 | 4 | 4 | n/a |
| Sensory fanout | 6 | 6 | 6 | n/a |
| Rec. com. synapses | 4 | 4 | 4 | n/a |
| Motor fan-in | 6 | 6 | 6 | n/a |

Table 10: **Training hyperparameters:** Miscellaneous parameters that were shared across all model types.

| Param Name | Param Value |
| --- | --- |
| Training epochs | 300 (150 for Liquid 9Hz) |
| Validation split | 0.05 |
| Optimizer | ADAM |
| Data shift | 16 |
| Data stride | 1 |
| Sequence length | 64 |
| Batch size | 32 |

Table 11: **Model sizes:** Number of trainable parameters in the recurrent part of the networks and the number of parameters of the CNN backbone.

| Model Type | Number of trainable parameters |
| --- | --- |
| Liquid | 12.3k |
| Liquid 3Hz | 12.3k |
| Liquid 9Hz | 12.3k |
| LSTM | 308.0k |
| Shared CNN Backbone | 103.7k |

## D.2 Performance repeatability

We compare the performance of 5 instances of training of the each model on the PyBullet dataset. We test the checkpoints on a single target test where the initial placement of the target is one of the four corners of the frame to maximize difficulty and evaluate robustness. The success rate over 100 trials for each is presented in Table 13.

The Liquid model averages a 89% success rate with a 11.4% standard deviation. The lowest performing instance succeeds three quarters of the time whereas the best achieve perfect performance. This shows that the model's hyperparameter tuning leads to consistently good checkpoints although training initialisation and stochastic optimization do, as in most learned models, have an impact on an instance's performance. Compared to the Liquid 3Hz and 9Hz, we see that irregular time-sampling with the Liquid 1-10Hz model leads to higher repeatability over multiple trials. LSTM does consistently well in-distribution over all training iterations.

Table 12: **CNN Architecture details:** Shared architecture of the CNN backbone that processed visual inputs for the different recurrent neural architectures.

| Layer Type | Input Dim. | Filters | Kernel | Stride | Activation |
|---|---|---|---|---|---|
| 2D Conv. | $144 \times 256 \times 3$ | 24 | 5x5 | 2 | relu |
| 2D Conv. | 70x126x24 | 36 | 5x5 | 2 | relu |
| 2D Conv. | 33x61x36 | 48 | 5x5 | 2 | relu |
| 2D Conv. | 15x29x48 | 64 | 3x3 | 1 | relu |
| 2D Conv. | 13x27x64 | 16 | 3x3 | 2 | relu |
| Flatten | 128 | n/a | n/a | n/a | linear |

Table 13: Repeatability tests: Success rate of target-in-corner test for five training runs ($N = 100$).

| Training Instance | Liquid | Liquid 3Hz | Liquid 9Hz | LSTM |
|---|---|---|---|---|
| 1 | 1.00 | 0.75 | 0.00 | 1.00 |
| 2 | 0.73 | 0.00 | 0.91 | 1.00 |
| 3 | 0.89 | 0.97 | 1.00 | 1.00 |
| 4 | 1.00 | 1.00 | 0.00 | 0.80 |
| 5 | 0.84 | 1.00 | 0.52 | 0.96 |
| **Average** | 0.892 | 0.744 | 0.486 | 0.95 |

