# OpenReview forum: "Gaussian Splatting to Real World Flight Navigation Transfer with Liquid Networks"
_robot-learning.org/CoRL/2024/Conference — CoRL 2024_

### Official Review · Reviewer_wnWZ · 2024-07-20
**Solid work, unsure about significance**

**Originality:** 2
**Technical Quality:** 3
**Clarity Of Presentation:** 4
**Potential Impact:** 2
**Recommendation:** 3
**Confidence:** 3

**Review:**

The paper is relatively well-written. One can tell from the amount of technical details given in the main text and appendices that the authors put much work in this submission to make it a solid research paper. Though this submission is mostly a combination of existing toolboxes (GS, NeuralODE), and the combination is rather "taking off-the-shelf" and simplistic, the successful use of GS and NeuralODE might be informative to peer researchers and practitioners. Other strengths include the thoughtful design of the dataset & augmentation scheme, including collecting data on reaching one target and testing on research multiple targets consecutively.

Weaknesses:
1. As mentioned, all existing methods and tools are used almost off-the-shelf, and the final system can be seen as simple, affecting the significance of the contribution.
2. The baselines feel quite weak -- see rebuttal questions below
3. The authors mention the method can generalize to outdoor scene even if only trained only in completely different indoor environment. This might actually be an indicator that realistic rendering is not that important anyways and maybe random real background might suffice. Authors are encouraged to think about more analysis on evaluating the usefulness of the background.
4. The task is rather simple, and there is only one task. One might be concerned how this framework works on more challenging tasks. In this simple task of reaching to red/blue target, a reasonable guess is that background, realistic or not, will not be that important in learning anyways (connects to my point 3)
5. Limitations are not discussed properly. There is only a section in Appendix discussing current failure cases, but for Limitations it should be much broader. Also the failure case section does not provide visuals (image/video).

**Quality Of The Limitations Section:**

1

**Questions For Rebuttal:**

- Can you design a baseline that takes timestep as input? Currently the LSTM baseline seems unfair since it cannot take the delta t as input. One simplistic baseline could be using a transformer, and for each frame of image you append a positional embedding using current t or delta_t.

- For a naive baseline of the Gaussian splatting, what about using random real indoor images as background instead of GS-rendered images? How will that affect outdoor performance? Further, what about putting the synthesized trajectories into high-quality renderers with environments, such as Habitat, or OmniGibson - that is, do we really need GS built from real images for this simple task? (This is different from Appendix C.2)

- What's the algorithm for imitation learning, just standard supervised learning (Behavior Cloning)? Please provide details of BC if tricks were necessary to avoid accumulative error and drifting.

- Do you really need physics simulation? Are you just using it to integrate the drone dynamics, which can be simply done by writing your own numerical integrator?

**Robotics Focus:**

4

**Summary Of Paper:**

The authors built a Gaussian Splatting reconstruction for an indoor lab, and use this GS to render the augmented trajectories (hard to collect such imitation trajectory in real world) so that imitation learning policy can be more robust due to seen more diverse data. They also adopted NeuralODE to deal with non-constant time step.

**Summary Of Recommendation:**

More baselines and ablations ought to be done to gain a better understanding where this good performance and zero-shot generalization really comes from. Also the paper should showcase its effectiveness in more than one task and more environments. But as mentioned, the paper, with some revision, might be useful to other researchers.

---

### Official Review · Reviewer_iDUo · 2024-07-21
**Real-world simulator with Gaussian Splatting for Visual Rendering and Flight Simulator for dynamics for flight navigation**

**Originality:** 4
**Technical Quality:** 4
**Clarity Of Presentation:** 4
**Potential Impact:** 3
**Recommendation:** 3
**Confidence:** 4

**Review:**

**Pros**

- Using Gaussian spatting allows task-specific and scene-specific photorealistic real-scene rendering, eliminating the sim2real gap visually. For flights, the flight dynamics simulator provides an easy expert data generator.

- By utilizing liquid neural networks, the policy can be trained on irregularly sampled data temporally, so that the network can learn to generalize across different frequencies, which is very useful and critical for real-time policy for dynamic flight navigation.

- The authors report task performance, ablating network design (liquid networks vs LSTM), and training data frequency (3Hz, 9Hz, non-constant sampling rates), and show that for GS-to-GS/Real settings, the proposed method (non-constant sampling + liquid network) achieves the best performance.

- Contains real-world experiment results with video visualization in the supplement video. Nice supplemental video!

**Cons**

- Missing limitations section.

- While the result of the proposed method is statically better than the baselines compared, the final absolute success rate for GS-to-Real of the proposed method seems a bit low given the task complexity (0.5). Any intuition for this number?

- There is no apple-to-apple ablation studies that ablates the performance effects coming from the photorealistic simulator, data sampling rate trained, or policy network choice. Thus it's hard to figure out which out of these system components are the most critical.
- It’s not clear how the current method would generalize (if so, how well) to dynamic scenes which is typical and important for flight navigation.

- The alignment of the GS scale with the simulator is manual, which is costly to easily scale to different task scenes and environments.

- To really make the case for the effects for GS for flight simulator rendering, the experiment of Sim-to-Real would be needed for a more apple-to-apple comparison.

- The paper only has one type of task (flight navigation with approach and turn), thus it's unclear how the proposed simulator scheme generalizes to more flight tasks.


**Questions**

- The authors assume the task is composed of two stages, approach and turn phases. While these make sense and seem to work well for the flight navigation task this paper considers, curious about the generalizability of these decomposed phases to more complex flight navigation tasks that involve more cluttered environments that require tighter maneuvers and significant pitch and roll motions. It would be helpful for the authors to provide some discussion on this potential limitation.

**Quality Of The Limitations Section:**

1

**Questions For Rebuttal:**

- The authors assume the task is composed of two stages, approach and turn phases. While these make sense and seem to work well for the flight navigation task this paper considers, curious about the generalizability of these decomposed phases to more complex flight navigation tasks that involve more cluttered environments that require tighter maneuvers and significant pitch and roll motions. It would be helpful for the authors to provide some discussion on this potential limitation.

- If time allows, to really make the case for the effects for GS for flight simulator rendering, the experiment of Sim-to-Real would be needed for a more apple-to-apple comparison.

- More visualization of the predicted and executed flight path in real world would be helpful

**Robotics Focus:**

4

**Summary Of Paper:**

The authors proposed a simulator scheme that provides photorealistic rendering from Gaussian Splatting with high-fidelity flight dynamics to train imitation learning policies for real-world flight navigation. They utilized liquid neural networks with irregularly time-sampled data training to increase policy robustness. Quantitative real-world flight tests show the trained policies from the proposed simulator effectively transfer to real-world flight navigation in a zero-shot manner.

**Summary Of Recommendation:**

Creative and smart integration of existing techniques from multiple fields: model architecture, neural scene reconstruction, and flight dynamics simulator, to leverage the best from different worlds to allow a higher-fidelity flight simulator with a policy model able to train from it. Idea for integration is great, but lacks extensive apple-to-apple ablations, and task diversity to make a confident prediction about the scalability of the method for more complex flight navigation scenarios.

---

### Official Review · Reviewer_5nTe · 2024-07-21
**Review of Gaussian Splatting to Real World Flight Navigation Transfer with Liquid Networks**

**Originality:** 4
**Technical Quality:** 4
**Clarity Of Presentation:** 3
**Potential Impact:** 3
**Recommendation:** 3
**Confidence:** 4

**Review:**

### Overview of review

Overall, I consider this to be among the better papers I have reviewed for CoRL and has a good coverage with contributions, experimentation and related discussion. The core ideas are rational and clearly stated, the experiments are sufficiently thorough and demonstrate the efficacy of the proposed methods, and the discussion, for the most part, helps readers digest the most salient points of the work.

### Strengths

- Clear ideas, well structured presentation and relatively strong contributions.
- There is a logical progression of experiments which successfully demonstrates the soundness of the basic policy training procedure, followed by the Gaussian splatting augmentation and then finally the sim-to-real transference both in and out of distribution. This helps build confidence at each stage that the presented rationalizations correspond quantitatively with evaluation metrics.
- Demonstration with real hardware.
- While not discussed in the main body of the paper, the additional information in the appendices on environment and training designs offers useful insight and context of how the training pipeline works.
- This work is relatively novel - as the authors note, it is not the first to consider the problem of training control policies in simulated photoreal environments constructed radiance fields, but to my knowledge, an exploration of the application of liquid neural nets in such a problem is indeed novel. In a circumstance less constrained for space, I might have wanted to see more comparisons with different radiance field techniques, but I do not think that is strictly necessary to demonstrate the core utility of the training pipeline being discussed here - as I am reasonably convinced that the proposed liquid network-based control would be able to generalize similarly well even when trained with other forms of photorealistic simulation, given that there are not too many strange rendering artifacts (though that would probably trip up most contemporary methods).

### Weaknesses

- I had originally intended to call into question the repeatability of the methods presented here but It is evident from the appendices that the design decisions and hyperparameters selections were made based on multiple tests. The appendices do not however numerically quantify how performant each of the 'best' hyperparameters were, relative to others considered, nor how well-performing the 'best-performing' checkpoints from the different training runs were relative to other checkpoints from other runs. If there are reportable values to back the design choices made, they would help strengthen the paper as an independent work and would not rely on readers to trust in the general effectiveness of each of the component methods used.

- Table 1 indicates that the 100 test runs are split 53 red and 47 blue. I do not believe it is ever clearly addressed why the split is not 50/50. I suspect it has something to do with the stated biases that some methods have to different colors of targets, but I can't be sure. Unless I missed something, that aspect of the training procedure could stand to be more clearly explained.

- There is no clearly separated discussion on the limitations posed by the proposed methods. The appendices discuss some of the failure cases but the main body of the paper does not do much to address the method's major failure modes and bigger-picture limitations and what sort of paths future work may take to address them.

**Quality Of The Limitations Section:**

1

**Questions For Rebuttal:**

1. Perhaps this is ignorance born from a limitation in my understanding of liquid networks, but it is surprising to me that the 9 Hz network performs poorly w.r.t. the 3 Hz network. I would have assumed that more rapid sampling might offer more opportunities for correction and therefore improved performance. Can the authors provide insight into why the faster sampling might contribute to worse performance? Based on appendix Fig. 8, it would seem that the dataset sampled for the 1 - 10 Hz main Liquid net has collectively more data that would exceed 3 Hz, so I would have expected the performance to skew closer to that of the 9 Hz net rather than 3 Hz. Evidently the data variation helps the main liquid net but I have not satisfactorily rationalized why the 9 Hz net would be worse. Would a 1 Hz net be better by extension?

2. How important are the recovery trajectories when constructing the training dataset? Based on the context of the paper, it seems like it would be important but it lacks a quantitative metric.

**Robotics Focus:**

4

**Summary Of Paper:**

The authors seek to address sim-to-real navigation policy transfer through the introduction of a simulator augmented for photorealism with Gaussian splatting and leveraging the out-of-distribution adaptability of liquid networks. The authors demonstrate the efficacy of their proposed imitation learning pipeline both in simulation and with real-world tests.

**Summary Of Recommendation:**

Overall, I believe this is a good paper and I would vote in favor of its acceptance. There are a few details however where I'm not entirely clear on the paper's and authors' positions, thus limiting my recommendation to a weak accept for now.

---

### Author Rebuttal · Authors · 2024-08-13

We thank the reviewers for their valuable input through this rebuttal process. We hope to have satisfied their concerns and to have provided sufficient clarity to their questions. We have taken into account their remarks in this version of the manuscript namely:
- A comprehensive limitations section
- Ablations and tests pertaining to:
     1. The repeatability of training (will contain all models for the camera ready submission)
     2. The importance of recovery maximisation in dataset generation
     3. The influence of irregularly time sampled data on LSTM vs Liquid
     4. A visualisation of the real world trajectory during a multi-step hike
- Clarification specific points in the text where requested (data split, behaviour cloning)

The changes are tracked in blue text colour in the revised version of the paper attached.

---

### Decision · Program_Chairs · 2024-09-04

**Decision:**

Accept

**Comment:**

Strengths:
1. Demonstration on real hardware, clear ideas and very well structured presentation. Good work on providing a nice appendix with details and supplementary video
2. Connecting sim2real work to liquid neural networks is very novel and promising

Weaknesses:
1. Discuss limitations of the approach more thoroughly - this is a major weakness appearing in every review.
2. More ablation can be useful in understanding which part of the contribution is most critical
3. Study is done only on one type of task, would others work?
4. is it possible to generalize to more task types and out door / dynamic scenes?

Reviewer 5nTe was concerned about the lack of quantitative metrics on hyperparameter choices and repeatability. The authors addressed this by clarifying their selection process and committing to a systematic evaluation in the final version. The reviewer also inquired about the importance of recovery trajectories, which the authors acknowledged and are currently investigating quantitatively. Lastly, the reviewer was curious about the impact of higher sampling rates, and the authors explained the potential downsides, such as overfitting.

Reviewer iDUo pointed out the lack of ablation studies and questioned the generalizability to dynamic scenes and complex tasks. In response, the authors clarified the existing ablations and discussed the limitations in handling dynamic scenes, along with potential future directions. The reviewer also raised a concern about the manual alignment of the Gaussian Splatting scale, which the authors acknowledged as a limitation but also highlighted the trade-offs with its benefits.

Reviewer wnWZ considered the combination of existing methods simplistic and the baselines weak. They also questioned the necessity of realistic rendering and physics simulation. The authors defended their contribution as a complete training scheme and are working on stronger baselines. They emphasized the importance of realistic rendering for complex tasks and highlighted the advantages of Gaussian Splatting for creating customizable simulators.

All reviewers suggest accepting the paper unanimously post rebuttal.